# Exchange biased anomalous Hall effect driven by frustration in a magnetic kagome lattice

Ella Lachman[1,2]*, Ryan A. Murphy [3], Nikola Maksimovic[1,2], Robert Kealhofer[1,2], Shannon Haley[1,2], Ross D. McDonald [4], Jeffrey R. Long [2,3,5] & James G. Analytis [1,2]

$Co_3Sn_2S_2$ is a ferromagnetic Weyl semimetal that has been the subject of intense scientific interest due to its large anomalous Hall effect. We show that the coupling of this material's topological properties to its magnetic texture leads to a strongly exchange biased anomalous Hall effect. We argue that this is likely caused by the coexistence of ferromagnetism and geometric frustration intrinsic to the kagome network of magnetic ions, giving rise to spin-glass behavior and an exchange bias.

[1] Department of Physics, University of California, Berkeley, CA 94720, USA. [2] Materials Sciences Division, Lawrence Berkeley National Laboratory, Berkeley, CA 94720, USA. [3] Department of Chemistry, University of California, Berkeley, CA 94720, USA. [4] Los Alamos National Laboratory, Los Alamos, NM 87545, USA. [5] Department of Chemical and Biomolecular Engineering, University of California, Berkeley, CA 94720, USA. *email: ellal@berkeley.edu

Magnetic Weyl semimetals (WSM) are predicted to host the Quantum anomalous Hall effect (AHE) at higher temperatures than magnetically doped topological insulators[1], and are therefore of substantial interest for spintronics technologies. One such material is $Co_3Sn_2S_2$, which has been the subject of intense research interest because the interplay of topology and magnetic order leads to a giant AHE in the presence of a weak ordered moment[2,3]. $Co_3Sn_2S_2$ is a shandite material, where the Co atoms form layers of 2D kagome lattice. S and Sn atoms are interleaved in the layers, with another Sn species in between the Co-S layers. $Co_3Sn_2S_2$ is a half-metallic material with only one component of spin contributing to the conductivity in the ferromagnetic state below $T_c = 175$ K[4], with each Co spin contributing ∼0.33 $\mu_B$ according to first principle calculations[3]. Recent calculations and measurements including ARPES and STM show that $Co_3Sn_2S_2$ is a WSM, with Weyl points located ∼60 meV above the Fermi energy[5,6]. The magnetic order of this material is not straight forward, and several studies have recently suggested $Co_3Sn_2S_2$ hosts a complex magnetic texture. Understanding how this magnetic complexity affects the topological properties of $Co_3Sn_2S_2$, and specifically the AHE, is the focus of this study.

Exchange Bias (EB) plays an important role in magnetic memory technologies, ensuring stability and protecting against volatility. Typically, it is the result of exchange interaction at the interface of a ferromagnet (FM) and another magnetic phase, often an anti-ferromagnet (AFM)[7]. As a result of this interaction, the EB effect manifests as a field-shift of the magnetic hysteresis loop. Defining $H_{C\pm}$ as the field at which the magnetization

changes sign along an $M(H)$ curve, the exchange bias can be parametrized as $H_{EB} = -(H_{C-} + H_{C+})/2$. This shift is related to the strength of the pinning exchange interaction between the FM and AFM[8].

Here we show that the coexistence of two magnetic phases in $Co_3Sn_2S_2$ leads to an exchange bias that strongly influences the AHE. In addition to being of fundamental interest, this opens the possibility of applying this mechanism as a basis for topological spin valve technologies.

## Results

**Low temperature anomalous Hall effect and exchange bias.** Single crystals of $Co_3Sn_2S_2$ were grown using the flux method and were either used pristine or silver epoxy contacts were attached for transport measurements. A plot of resistance as a function of temperature presented in Fig. 1a, b indeed shows a magnetic transition at a temperature of 175 K.

The magnetic easy axis in $Co_3Sn_2S_2$ is perpendicular to the Co kagome planes, and parallel to the crystal's c-axis. The system exhibits magnetic hysteresis in both magnetization and Hall resistance when sweeping a perpendicular magnetic field at temperatures below the magnetic transition.

At a temperature of 2 K, the magnetic transition appears as a sharp step in $R_{xy}$ at $\mu_0 H_c = 115 \pm 10$ mT (Fig. 1c). When accounting for mixing of the longitudinal resistance (see methods), the resulting hysteresis loop is characteristic of the AHE with a crucial difference: The loop is not centered around zero applied field, with the offset depending upon the thermal

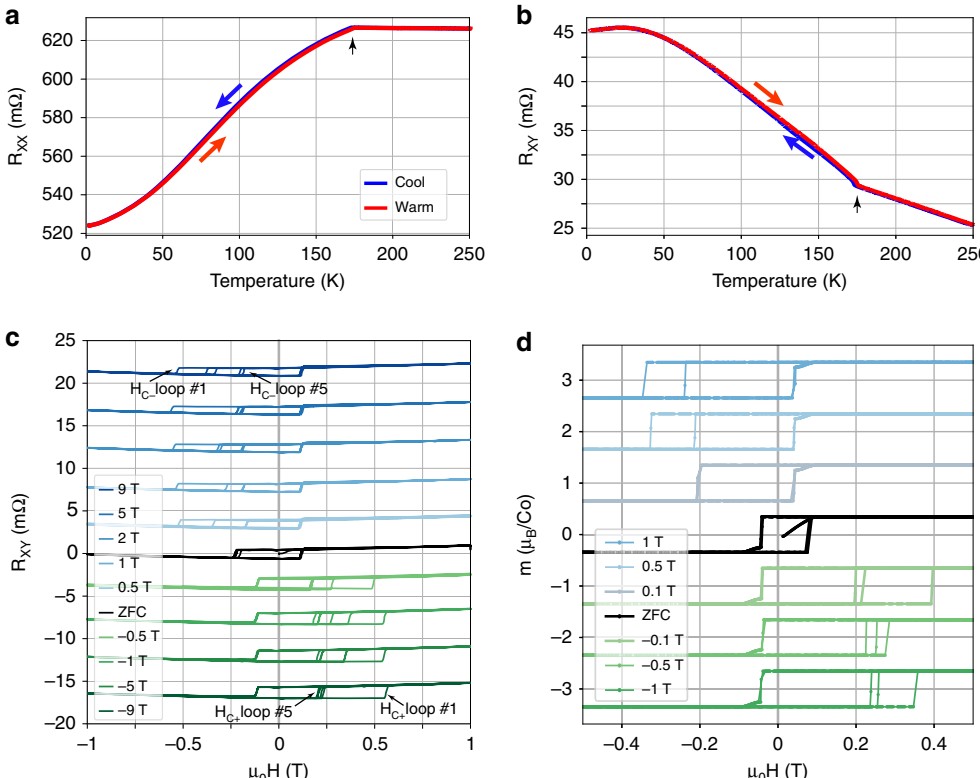

**Fig. 1 Magnetic transition and low temperature behavior of $Co_3Sn_2S_2$ single crystal sample. a**, **b** Longitudinal ($R_{xx}$) (**a**) and transverse ($R_{xy}$) (**b**) resistance as a function of temperature shows a magnetic phase transition at 175 K as evident by the change in slope (marked with black arrow). **c** Exchange biased AHE. Hall resistance as a function of magnetic field at a temperature of 2 K. The field is swept from between +1 T and −1 T and the sweep is repeated 5 times, resulting in a different $H_{C-}$ ($H_{C+}$) for cooling down in positive (negative) magnetic fields. The coercive fields on the opposite side are the same and therefore fall on top of each other in the plot. **d** Exchange biased magnetization at 2 K, showing similar EB effect to the one in transport. The initial field sweep seems to be of lower $H_C$ due to the slower field sweep rate in this measurement (see SI). Curves in **c**, **d** are shifted vertically for clarity and an easier comparison.

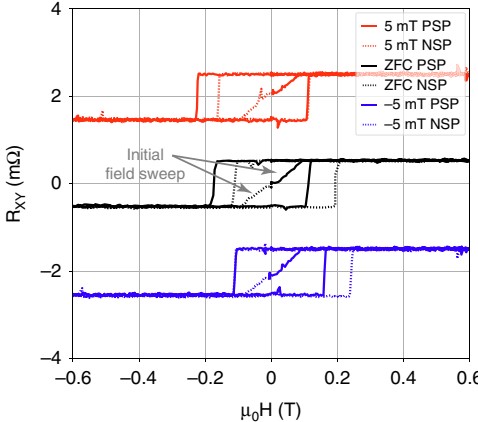

**Fig. 2 Spontaneous exchange bias.** $R_{xy}$ as a function of applied magnetic field for different low values of cooling fields at 2 K. For each field, both a positive sweep protocol (PSP) and a negative sweep protocol (NSP) were performed, where the sample is warmed up to 250 K—well above $T_C = 175$ K—between the two to negate the SEB magnetization effect. The initial field sweeps for the zero-field cooled curve are marked with arrows, but are clear inside the hysteresis loop area for all fields. These indicate that the initial magnetic state is unsaturated, and that the SEB is indeed induced at the low temperature, by the specifics of the field sweep protocol. The curves for different cooling fields are shifted vertically for clarity and an easier comparison.

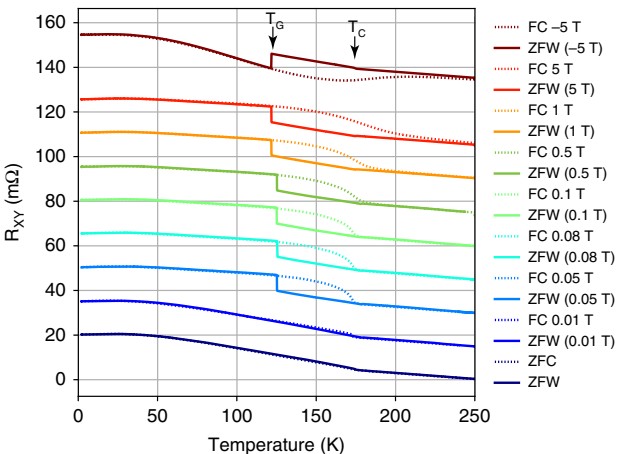

**Fig. 3 A transition at 125 K.** $R_{xy}$ as function of temperature for zero-field warming (ZFW) up after in-field cooling (FC) of the sample. For each cooling in-field (dotted lines) the zero-field warming up (solid) is of the same color. On the zero-field warming up curves, the transition at 125 K is evident as a jump to a different value. For cooling in positive (negative) field, the jump at 125 K is to a lower (higher) value. The curves for different fields are shifted vertically for clarity and an easier comparison.

and magnetic history. The transition at positive applied fields is at $\mu_0 H_c = 115$ mT, but on the negative part it is between $\mu_0 H = -200$ mT and $\mu_0 H = -230$ mT and changes between sequential repeated field sweeps. When cooling the sample in the presence of magnetic fields, a negative field shifts the positive coercive field as far as 550 mT initially, and to 200 mT after relaxation with repeated field sweeps. Positive fields shift the negative coercive field to similar fields with opposite sign. Analogous effects are also evident in magnetization measurements performed on a different sample at the same temperature (Fig. 1d). The asymmetry of the hysteresis loop and its dependence on magnetic history is the signature of EB[9], and serves as evidence that the interpretation of the magnetic phase of $Co_3Sn_2S_2$ as a simple FM phase in which the Co spins point out of plane is inadequate.

Remarkably, even when cooling the sample with no magnetic field applied (zero-field cool (ZFC)) a small EB appears (see Fig. 2, this can also be seen in Fig. 1c). Though first attributed to a possible remnant field in the superconducting magnet, further investigation shows this EB to be a spontaneous one[10]. The spontaneous EB (SEB) can be isothermally induced by the initial field sweep direction at low temperatures, and demonstrates the importance of the thermal and magnetic history, and hence the experimental protocol. Figure 2 shows magnetic field sweeps at 2 K, with two different field sweep protocols. One is "positive" where the field is swept $0 \rightarrow +1T \rightarrow -1T \rightarrow +1T$, and the other is a "negative" sweep protocol, where the field is swept $0 \rightarrow -1T \rightarrow +1T \rightarrow -1T$. These two sweeps were each performed after zero-field cooling and after in-field cooling. Let us first analyze the zero-field cooled sweeps. The positive sweep protocol (PSP) results in $H_{EB} = 31.5$ mT, and the negative sweep protocol (NSP) yields $H_{EB} = -40$ mT. Both positive and negative protocols produced the same saturation value of $\Delta R_{xy} = 1$ mΩ, and show a similar magnitude of $H_{EB}$. In addition, the initial value of $R_{xy}$ for both field sweeps is close to zero, indicating that the initial state of the material is unmagnetized or of very low

magnetization. This rules out the naive explanation of remnant field in our magnet. When the sample is cooled in a small field, the SEB effect and the normal EB effect combine to form the total EB. This is evident, for example, in the 5 mT cooling PSP, in which $|H_{C-}|$ is larger than that of the NSP. The same applies for the −5 mT cooling, but now it is the NSP in which $H_{C+}$ is larger than that of the PSP. The presence of EB and SEB at low temperatures suggests the interaction of the FM moment in $Co_3Sn_2S_2$ with an additional AFM or a spin glass (SG)[7,11].

**Anomalous Hall and magnetism at intermediate temperatures.** The sample's resistance as a function of temperature in Fig. 1a, b exhibits no clear signature of an additional magnetic phase transition that may explain the appearance of EB. However, the existence of a transition near 125 K becomes readily apparent in $R_{xy}$ when thermaly cycling after field cooling, as shown in Fig. 3. In this measurement protocol, a field was applied while cooling (where FC data is collected), swept to zero at 2 K and then the sample was warmed up with no field to 250 K (ZFW data collected). For cooling fields $|\mu_0 H_{cool}| > 0.05$ T, a clear transition can be observed at 125 K, marked by a sudden change in $R_{xy}$ in the ZFW curves. We denote this temperature as $T_G$. This demonstrates there is an interaction between the FM moment and another phase, which pins the zero-field value of $R_{xy}$ below that temperature. FC and ZFW curves merge at the FM transition at 175 K. This suggests that the phase below $T_G$ retains the memory of the cooling conditions, and forces a change in the FM ordering at zero field.

In order to look for clues as to the origin of the anomaly, magnetization and transport measurements as a function of magnetic field were performed at different temperatures around $T_G$ for samples cooled in a field $\mu_0 H_{cool} = 0.5$ T (Fig. 4a, b respectively). These hysteresis loops reveal a qualitative difference between low temperatures and higher temperatures still below the Curie temperature of 175 K. The EB is diminished at $T_G$, though the system is still magnetic. This is clear from the different high field response at positive and negative fields, as well as from the magnetic hysteresis. This hysteresis no longer has the typical square shape of the AHE (transport) or a ferromagnet

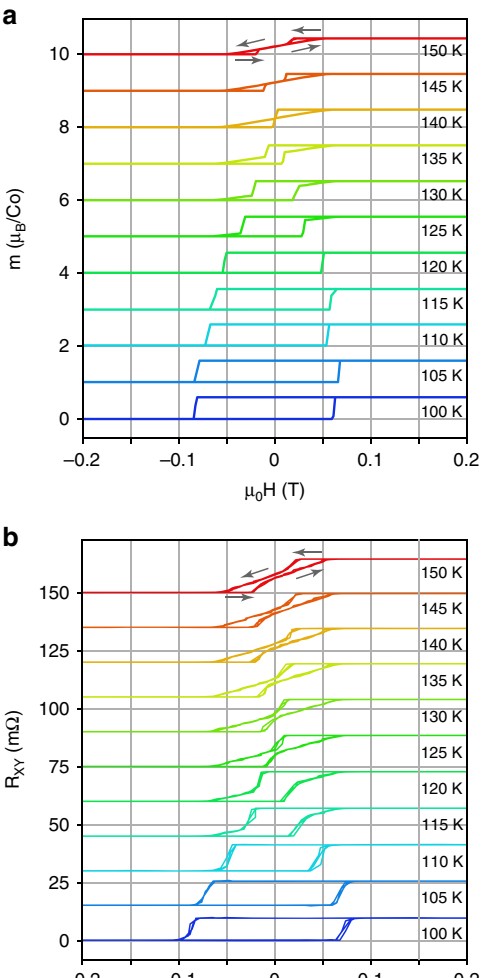

**Fig. 4 Magnetic hysteresis at intermediate temperatures with field parallel to the c-axis.** The samples were cooled in an applied field $\mu_0 H = 0.5$ T showing the magnetic hysteresis loops for temperatures near 125 K. **a** Magnetization as a function of applied magnetic field. The square shape of the loop at 100 K gradually changes to a bipartite transition. **b** $R_{xy}$ as a function of applied magnetic field. The square, AHE characteristic shape at 100 K gradually changes to a bipartite transition with almost no hysteresis at zero field. The field value of the hysteresis step continuously evolves from positive to negative field upon lowering temperature with the temperature at which it crosses zero field corresponding to the step in the AHE seen in Fig. 3.

(magnetization), which suggests an unusual energy landscape for the phase underlying the EB and the ferromagnetism (see SI).

The evolution of the magnetic hysteresis in $R_{xy}$ as the temperature is increased through $T_G$ sheds light on the 125 K anomaly presented in Fig. 3. When the sample is cooled in field $|\mu_0 H_{cool}| > 0.05$ T to low temperatures ($T < T_G$) it is magnetized, and retains this magnetization as the field is swept to zero. When subsequently warming up in zero applied field, the value of $R_{xy}$ retains its magnetized value, until $R_{xy}$ at zero field becomes significantly lower (about 80%) than its high field value at the same temperature.

The appearance of EB, together with the unconventional shape of the magnetization hysteresis loops indicates the coexistence and interaction of two magnetic states. These observations also raise the interesting possibility that the same spin system provides

both the uncompensated FM moment and the correlations that conspire to give the AHE an EB below 125 K.

**Magnetic and thermodynamic properties of the 125 K phase.** A recent work using muon spin rotation[12] concluded that the magnetism in $Co_3Sn_2S_2$ originates solely from the Co atoms, and that there is a phase transition from a higher temperature in-plane AFM to a low temperature out-of-plane FM at $\sim 90$ K. This is not entirely consistent with our observation of EB, as EB cannot arise from a simple FM phase with no other interactions. To resolve this, we measured the magnetic moment of a $Co_3Sn_2S_2$ crystal as a function of temperature in both out-of-plane (Fig. 5a) and in-plane (Fig. 5b) directions. In the out-of-plane direction, a ZFC magnetization measurement while warming up in a 10 mT field shows the expected magnetic transition at 175 K, with an additional peak at 125 K. The same measurement protocol applied in the in-plane direction reveals an AFM-like cusp at 175 K, followed by a rise in magnetization at 125 K. The moment amplitude in the ab plane is two orders of magnitude lower than the out-of-plane moment. This is in agreement with the AHE appearing in transport, as well as with other magnetic measurements done previously[2,3,13]. The work by Kassem et al.[13] includes measurements of magnetization and AC susceptibility, which were interpreted as indicating an anomalous magnetic phase preceding the FM phase when cooling. Here, EB clearly shows that two types of magnetism coexist below 125 K. Heat capacity measurements were also performed in order to characterize the 125 K phase transition. The results of these measurements are presented in Fig. 5c. A clear transition appears at 175 K, but no significant feature is visible at 125 K, even when following the FC-ZFW protocol mentioned above that shows a transition in transport when cooling in fields $|\mu_0 H_{cool}| > 0.05$ T (as seen in Fig. 3). This is evidence that the 125 K transition does not correspond to an additional long range order, but of the freezing of a spin glass.

## Discussion

The classical method for creating EB in materials is to combine the FM material with an AFM one. The AFM is used as a pinning layer, as the exchange interaction at the FM-AFM interface pins the FM layer's magnetization, resulting in a higher coercive field needed to flip its magnetic orientation. It was later discovered that EB can also be induced by a combination of a FM phase with phases other than AFM[7], such as a ferrimagnet[14] or a spin glass (SG)[11]. The presence of exchange bias below $T_G = 125$ K, evidenced in Figs. 1 and 2, points to the coexistence of ferromagnetism with another phase. As we discuss below, we suggest this phase is a spin glass arising from strong magnetic frustration in the kagome lattice.

The first clue as to the nature of the coexisting phase is the unusual bow-tie like structure of the $M(H)$ sweeps in the range $T_G < T < T_c$, shown in Fig. 4. On sweeping the field up, the system begins with a linear response, saturating at a magnetization $M_0$, indicating the polarization of the moments. This is in contrast to an ordinary FM, where the moments remain polarized until a sufficiently negative field can flip the spins. In the present case, the system depolarizes at positive fields, regaining its paramagnetic response. As a minimal Landau model, this can be understood as the interplay of two terms in the free energy, one favoring a FM structure $M = \pm M_0$, and another with a minimum favoring $M = 0$. This could be the coexistence of an AFM, as claimed recently[12], or a spin glass.

The difficulty with reconciling an AFM coexistence at $T_G < T < T_c$, is that EB is not observed in this range, but rather at $T < T_G$. This means that the coexisting phase must become stiffer below $T_G$. However, the absence of a heat capacity anomaly

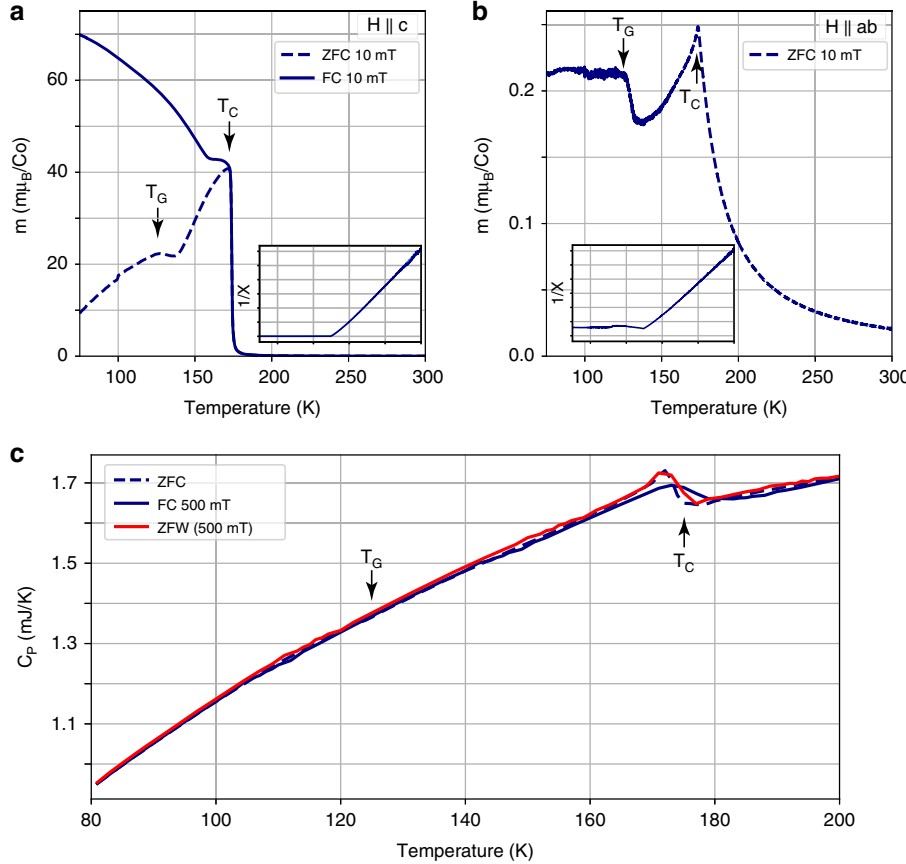

**Fig. 5 Magnetic and thermodynamic properties of the transition at $T_G$. a, b** In-plane and out-of-plane magnetization measurements on a single crystal. **a** Out-of-plane (H || c) magnetization as a function of temperature while warming up the sample after ZFC (dashed) and field cooling (solid) and measuring under a field of 10 mT. The curves are characteristic of a FM with a Curie temperature of 175 K, with a small feature at 125 K (marked with a small arrow on the ZFC curve). **b** In-plane (H || ab) magnetization as a function of temperature while warming up the sample after ZFC under a field of 10 mT. The curve is characteristic of an AFM with a Néel temperature of 175 K, with a feature at 125 K. Additional magnetization data can be found in Supplementary Note 3. **c** Single crystal heat capacity as a function of temperature for 80−200 K. These measurements reveal no feature at 125 K.

at $T_G$ suggests this does not freeze a significant fraction of the degrees of freedom, consistent with the muon measurements which have confirmed the absence of a competing long range order at low temperatures[12].

We suggest that a frustration-driven spin glass undergoes a freezing transition at $T_G$, resulting in a stiffening of the glass and the stabilization of a finite moment at zero applied magnetic field. This interpretation can explain all of the present observations including exchange bias, and an absent heat capacity anomaly[11]. This also explains the temperature dependence of the $M(H)$ curves. In the range $T_G < T < T_c$, the system is a FM coexisting but not interacting with a dynamic spin glass, leading to a ground state of the system with no net moment at zero field, a paramagnetic linear susceptibility at low fields, and saturation of the moments at sufficiently high fields. Below $T_G$, the spin glass freezes into a stiff phase, pinning the FM into one of two states with $\pm M_0$, causing the hysteresis loop to open up, shift from the origin, and take the familiar form of a more conventional FM system.

A SG would be expected to show time dependent effects like magnetic relaxation or magnetic memory. Due to the coexistence with the ferromagnetism, the signal arising from the SG relaxation is expected to be (and was indeed) too low to resolve in the conventional relaxation experiments. We have therefore performed a different relaxation test in which we pause the magnetic field sweep during the hysteresis loop. This is qualitatively similar to isothermal DC magnetization experiments used to probe glassy

dynamics. By polarizing the FM at high negative fields and pausing at low positive fields we are in the region where the FM is softer (close to $H_c$) and the dynamics of the SG may be revealed. In Fig. 6a such a measurement is presented, showing a relaxation time of several minutes. All temperatures near $T_G$ show relaxation on a long timescale, both above and below. This is in agreement with our description of a spin glass coexisting with the FM phase, and undergoing a freezing transition at $T_G$.

AC susceptibility measurements around $T_G$ presented in Fig. 6b, c reveal a small frequency dependence of the transition peak, indicating a glassy transition. The Mydosh parameter[15,16], used to qualitatively identify the type of glassy dynamics associated with a transition, is ≈0.0025 for $T_G$, within the range expected of a spin glass transition. As the existence of slow DC magnetization dynamics both below and above this feature suggests this glassy transition differs significantly from a classic freezing transition, we additionally probed $T_c = 175$ K. While the real part of the AC susceptibility is dominated by the FM response which does not show a significant frequency dependence, the imaginary part does. An estimate of the Mydosh parameter using the $\chi''$ signal is ≈0.006, also consistent with a spin glass. Therefore, glassy dynamics onset at a freezing transition of 175 K, giving rise to the complex energy landscape described above (see also Supplementary Note 1). The small difference between Mydosh parameters at $T_c$ and $T_G$ as well as the qualitatively similar DC relaxation behavior below and above $T_G$ are thus consistent with identifying the 125 K transition as a

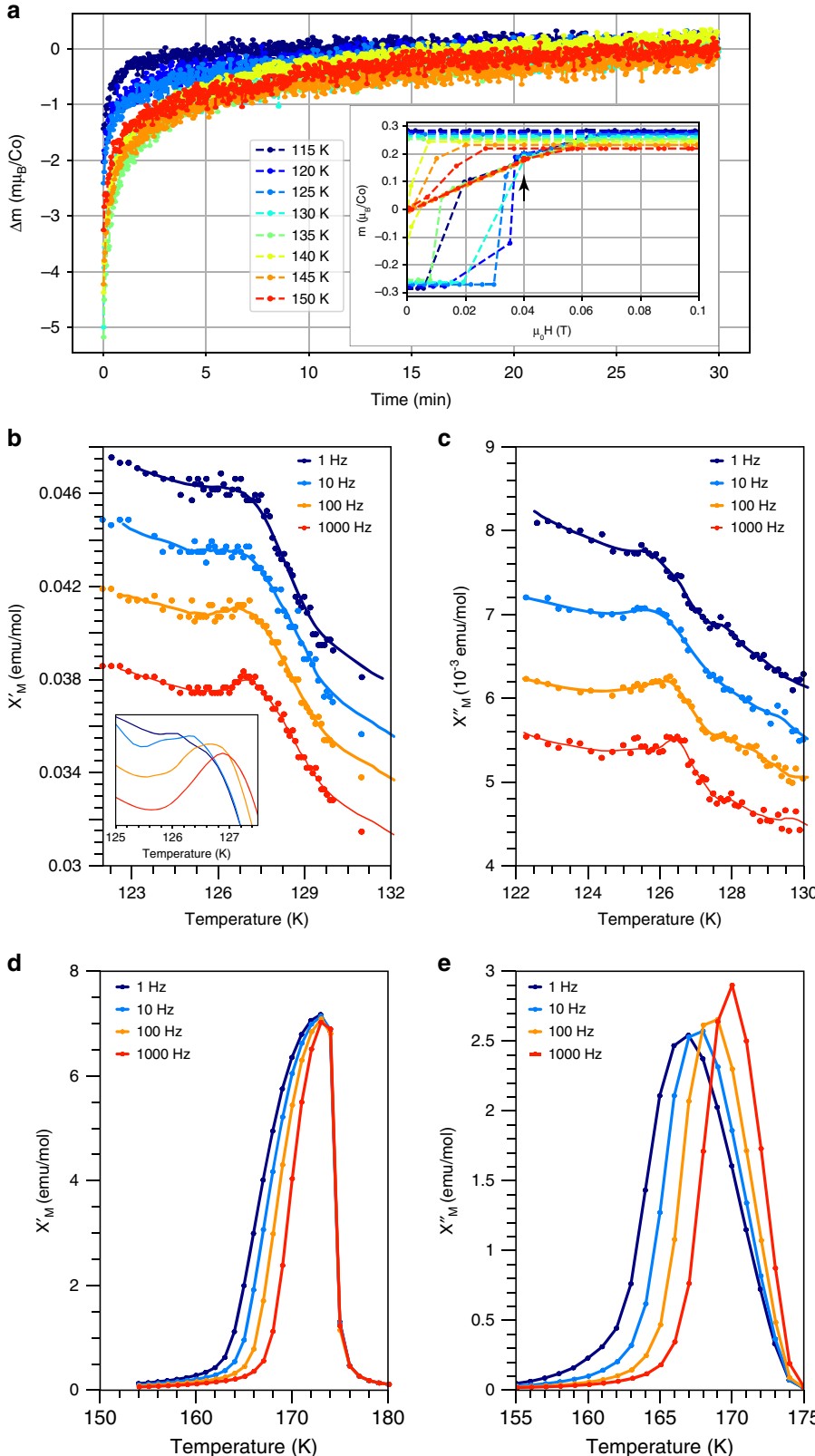

**Fig. 6 Non-trivial dynamics of magnetization and AC susceptibility measurements around $T_G$. a** Magnetic relaxation at different temperatures reveals a relaxation time on the order of minutes. The inset shows the point on the hysteresis loop where the field ramp was paused ($\mu_0 H = 40$ mT), and the relaxation measurement was taken (marked with an arrow). After a period of 30 min, the field sweep was resumed, saturating the magnetization to the maximum value for the specific temperature. **b**–**e** AC susceptibility measurements acquired with zero applied DC field and an oscillating field of 6.2 Oe, showing real and imaginary components of the signal at $T_G$ (**b**, **c**) and $T_c$ (**d**, **e**). Peak shifts by ~0.2 K per decade frequency at $T_G$ and 1 K at $T_c$. Smoothed curves are guides for the eye. $T_G$ curves have been vertically offset for clarity. Inset of **b** shows smoothed curves overlaid as a visual aid for the observed frequency dependence.

second freezing transition. At this transition the FM moments can be pinned by the spin glass, leading to the observed EB.

The spin glass is unlikely to be disorder driven. All studies of $Co_3Sn_2S_2$ have observed this magnetic transition to occur at the same temperature. This includes ours and others' samples that are clean enough to show quantum oscillations[2] (see Supplementary Fig. 6). The spin glass is therefore most likely driven by frustrated interactions on the kagome lattice. The observation of AFM correlations in ref. [12] using muon spectroscopy may be consistent with this interpretation, of frustrated in-plane AFM correlations causing weak in-plane canting. The interplay between the coexisting SG and the FM subsequently turns on at $T_G$, manifested as a second freezing transition leading to the exchange biased AHE presently observed.

$Co_3Sn_2S_2$ is a magnetic Weyl semimetal displaying exchange bias intrinsically, without the need for doping or layering. The exchange bias in this material can also be induced spontaneously at low temperatures and by low magnetic fields. We suggest that the origin of the exchange bias is the frustration in the kagome magnet, which leads to a ferromagnetic state that is simultaneously glassy below $T_c = 175$ K, leading to pinned moments below $T_G = 125$ K. The combination of these behaviours leads to exchange bias and spontaneous exchange bias that are strongly evident in the anomalous Hall effect. We emphasize that magnetism plays an important role in the robustness of the QAHE in magnetically doped topological insulators[17], and therefore may play a crucial role in unlocking the possibility of a QAHE in low-dimensional structures of $Co_3Sn_2S_2$. The interplay of magnetic frustration and topology in $Co_3Sn_2S_2$ provides an example of the potential utility of these materials for future spintronics technologies.

## Methods

**Single crystal growth**. Single crystals were grown from a stochiometric ratio of elements using the self-flux method (Sn flux). The elements were placed in $AlO_x$ crucible and sealed in an evacuated quartz tube.

**Transport and magnetization measurements**. Transport measurements were performed in a Quantum Design PPMS, with current flowing in the ab plane. Distance between current contacts was 1 mm, between $V_{xx}$ contacts was 0.3 mm, and between $V_{xy}$ contacts 0.4 mm as measured under an optical microscope. In all, 100 μm Pt wires were used. Mixing was accounted for by removing a constant ratio of $R_{xx}$ from $R_{xy}$ data for all temperatures and fields. This is purely a geometric factor as it is independent of the measurement conditions and is constant for all the transport measurements performed on a specific sample. The ratio was determined to remove symmetric contributions to $R_{xy}$ by leveling the high field values. This protocol was chosen because due to the non-symmetric nature of exchange bias, anti-symmetrising the data was not possible. Non-mixed data can be seen in Supplementary Note 6. Magnetization measurements were performed in a Quantum Design MPMS3 on a quartz rod.

**AC magnetic susceptibility measurements**. AC magnetic measurements were performed in a Quantum Design MPMS-XL SQUID magnetometer. In all, 33.5 mg of crystals were stacked using molten eicosane as adhesive, and were mounted in a flat bottomed quartz tube for measurements.

## Data availability

The datasets generated during and/or analysed during the current study are available from the corresponding author on reasonable request.

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

## Acknowledgements

This work was supported by the National Science Foundation under Grant No. 1607753. E.L is an Awardee of the Weizmann Institute of Science - National Postdoctoral Award Program for Advancing Women in Science. R.K. is supported by the National Science Foundation (NSF) Graduate Research Fellowship under Grant No. DGE-1106400. E.L., N.M. and S.H. acknowledge support from the Gordon and Betty Moore foundation's EPiQS Initiative through Grant GBMF9067. High field measurements were performed at the National High Magnetic Field Laboratory, which is supported by National Science Foundation Cooperative Agreement No. DMR-1157490 and the State of Florida. R.D.M was supported by Center for Advancement of Topological Semimetals, an Energy Frontier Research Center funded by the U.S. Department of Energy Office of Science, Office of Basic Energy Sciences, through the Ames Laboratory under contract DE-AC02-07CH11358. R.A.M and J.R.L. are supported by Department of Energy grant DE-SC0019356.

## Author contribution

E.L. performed the crystal growth, transport and magnetization measurements, as well as data analysis for the above measurements. R.A.M. and J.R.L. performed AC susceptibility measurements and interpreted these results. N.M., R.K., and S.H. performed high magnetic field measurements. E.L., R.D.M., and J.A. devised the experiments and interpreted the results. E.L. and J.A. wrote the paper with contributions from other authors.

## Competing interests

The authors declare no competing interests.
