## [Peer Review File · Nature Communications]

Reviewers' comments:

Reviewer #1 (Remarks to the Author):

This work reports exchange biased AHE in $\text{Co}_3\text{Sn}_2\text{S}_2$. The result looks very interesting and the data looks nice.

Having said that, the major concern is the lack of the evidence for spin glass state. The observed EB has been suggested to be due to the emergence of spin glass which coexist with the FM ordered phases. However, there is no critical evidence for spin glass. (there are some weak indirect evidences.) The author mentioned no relaxation behavior has been observed. Have the authors performed AC susceptibility? Given spin glass is directly related to the main claim of this manuscript, the clarification of it would be very important.

In addition, the current measurements appear not complete to support a solid claim, particularly when magnetism of this material system might be complicate. To make a complete story and to understand the unusual EB phenomena better, several other measurements will be needed. For example, extend the magnetization measurements (Fig. 3) to low temperature and compare with the Hall resistance (Fig. 1), magnetic field orientation dependence EB in magnetization measurements, etc. Further, for consistency the heat capacity should be performed under the same magnetic field as the magnetization in Figs. 5a-c.

Reviewer #2 (Remarks to the Author):

The manuscript focuses on the anomalous Hall effect in $\text{Co}_3\text{Sn}_2\text{S}_2$, a ferromagnetic Weyl semimetal which has been widely studied in the last two years. The manuscript reports on an exchange bias observed in the anomalous Hall effect which the authors use to support the claim, that in $\text{Co}_3\text{Sn}_2\text{S}_2$ there are two coexisting magnetic phases (ferromagnetism and (frozen) spin glass). They speculate that the coexistence of these two phases can give rise to an intrinsic (short range) exchange bias in the low temperature limit ($T < 125\text{K}$), where the spin glass phase is frozen. Similar results have been obtained by muon spin resonance suggesting a slightly different temperature window for the coexistence of an antiferromagnetic (not spin glass!) and ferromagnetic phase (90K-175K). The results in the present study do not corroborate this picture, as in this case one would expect to observe the exchange bias in this temperature window and not below.

To further underpin their findings and shine some light on the relevant mechanism they show the exchange bias to be present also in magnetization measurements in the low temperature region, while no exchange bias is present but an additional feature appears for $125\text{K} < T < 175\text{K}$. This, together with the exchange bias supports their claim of the two coexisting phases in this material also for the low temperature region, contradicting the findings of the muon spin resonance study. As such, I believe this study is an important contribution to the ongoing discussion on the exact magnetic phase diagram/magnetic state of $\text{Co}_3\text{Sn}_2\text{S}_2$. I recommend the manuscript for publication in Nature Communications after the following questions have been addressed:

Major questions:

1. Usually, the proof for the presence of a spin glass phase are non-trivial time dependent magnetization effects. The authors state that no such time/frequency dependent changes were observed also in the spin glass phase, which they claim is because it is a dense spin glass and the time scales are too low. However, in Ref. 12 a characteristic timescale of 1s is stated for the spin glass phase. How were the time dependent experiments performed (ac-susceptibility?). Any independent proof of the spin glass phase in the samples used for this study would extremely strengthen the point of the manuscript in my opinion.
2. In the study by Guguchia et al. (Ref. 11) the presence of an antiferromagnetic phase was shown to be present in the range from 90K to 175K. Is there any feature in the heat capacity down to

80K which might corroborate these findings? The authors show the heat capacity only down to 100K, so the temperature window should be extended down to at least 80K to strengthen their argument for the spin glass phase.

3. The authors show the low field Hall measurement in Fig. 1&2, then show temperature dependent magnetization measurements in Fig. 3, how do the two measurements relate? Is the EB the same for magnetization and Hall measurements also for different temperatures? Is the spontaneous EB also visible? How does the magnitude of the anomalous Hall effect compare to the size of the step in Fig. 4 at 125K (rough estimate from other data shows 50% of maximum AHE amplitude, but comparison is hard [see minor question 2])? Is this something to expect from the frozen spin glass picture? Can the fraction of the $\text{Co}_3\text{Sn}_2\text{S}_2$ which is pinned be estimated from this?

These questions should be addressed by also showing the Hall measurements for different temperatures in a similar fashion as in Fig. 3

Minor questions:

1. Since the authors say they cannot use the a symmetrization due to the symmetric nature of the exchange bias, they subtract a geometric factor from the transverse data. As such, for being able to reproduce the data/evaluation I think it is crucial that also the longitudinal transport data (R_{xx}/ρ_{xx}) are shown to be able to judge possible artifacts introduced by this approach.

2. To be able to compare the manuscript to other literature also regarding the effect magnitudes, the authors have to either convert their graphs to resistivities (not resistance) or give the sample size/contact spacings, so other scientists are able to convert the results to said units. Otherwise, it is very hard to compare the findings quantitatively.

3. Can the spontaneous EB also be related to domains?

Reviewer #3 (Remarks to the Author):

In the manuscript NCOMMS-19-23172, Lachman et al. reported the exchange bias effect observed in the ferromagnetic Weyl semimetal $\text{Co}_3\text{Sn}_2\text{S}_2$. Although the magnetic behaviors have been reported multiple times in literature, a clear picture of the magnetic order is absent. For this material, it is established that there is a strong magnetic anisotropy below T_c , which is known to be around 175 K. However, below this temperature, several studies have indicated the existence of a second magnetic transition and the possible existence of complex spin state or anomalous phase, for example, according to the result reported in the paper by Kasseem et al. [ref. 12]. Indeed more experimental work is necessary to clarify the magnetic order and its origin. In the current work, by the Hall effect and magnetization measurements under field-cooling and zero-field cooling conditions, the authors report the signature of exchange bias effect, which from a different angle demonstrates the existence of the anomalous phase. This result seems to indicate the coexistence of the ferromagnetic phase and the anomalous phase. To reconcile the observed phenomena, the author proposed a spin glass transition below $T_G = 125$ K, and assume the FM onsets with frustrated AFM correlations, which then freezes at T_G into a dense spin glass. Since the provided evidences are quite limited in this manuscript, I do not think they are convincing to support the proposed physical picture. I do not recommend it as a publication in Nature Communications. Below I list a few suggestion to further improve the manuscript for a publication at a less selective journal.

1. The authors define the additional phase transition as spin-glass transition (T_G) at 125 K. However, from ZFC magnetization curve (Fig. 5a), the transition anomaly seems to be around 140 K, while from the ZF magnetization curve (Fig. 5a), an anomalous temperature point seems to be around 160 K. Taking all the information together, the transition temperature at 125 K does not appear to be well defined. Additionally, it is debatable to relate the sharp jump of Hall resistivity at T_G (Fig. 4) to the anomalous magnetic transition. The sharp jump of the Hall resistivity should be directly related to the ferromagnetic domain dynamics, but it is not necessarily related to the spin-glass transition.

2. The provided experimental data is not enough to support the existence of dense spin glass and antiferromagnetic correlation above TG. These existence of these two, however, are the core ingredients of the proposed physics behind.

3. The information in the figure is not clearly conveyed. For example, The curves in fig. 1b, Fig. 2, Fig. 3, and Fig.4 seem to be vertically shifted for clarity, but the captions failed to indicate such information. Additionally, the temperature information for each MH is encouraged to be labeled, as there is enough space in the figure.

4. The authors claimed to deduce the transition temperature of 125 K from Fig. 3a. This is also debatable. The temperature evolution of the MH curves is too subtle. More solid evidence is required.

NCOMMS-19-23172 reviewers' comments

Contents

NCOMMS-19-23172 reviewers' comments	1
Reviewer #1 (Remarks to the Author):	2
Reviewer #2 (Remarks to the Author):	4
Reviewer #3 (Remarks to the Author):	8

Reviewer #1 (Remarks to the Author):

This work reports exchange biased AHE in Co₃Sn₂S₂. The result looks very interesting and the data looks nice.

We thank the reviewer for the positive comment. We agree that the results are very interesting.

Having said that, the major concern is the lack of the evidence for spin glass state. The observed EB has been suggested to be due to the emergence of spin glass which coexist with the FM ordered phases. However, there is no critical evidence for spin glass. (there are some weak indirect evidences.) The author mentioned no relaxation behavior has been observed. Have the authors performed AC susceptibility? Given spin glass is directly related to the main claim of this manuscript, the clarification of it would be very important.

The reviewer brings up an important point. A spin glass phase is usually characterized by magnetization relaxation, and by a frequency dependence in AC susceptibility. However, in our case the spin glass coexists with a strong ferromagnetic order with a higher transition temperature. As such, all signatures of the spin glass would be corrections compared to the signal from the ferromagnetic phase.

Having said that, we have performed a magnetization relaxation measurement where we pause the magnetic field sweep during the hysteresis loop. By polarizing the ferromagnet at high negative fields and pausing at low positive fields we are in the region where the FM is softer and indeed these measurements show a magnetization relaxation time of several minutes. This measurement is presented in Fig 6 (a). in the revised manuscript.

AC susceptibility measurements were also added to the revised manuscript, showing a peak in the vicinity of 125K and a frequency dependence of the peak location. This measurement is presented in Fig 6 (b,c) and is accompanied by this paragraph in the revised manuscript:

“AC susceptibility measurements around T_G presented in Fig. 6(b,c) reveal a small frequency dependence of the transition peak, indicating a glassy transition. The Mydosh parameter [15, 16], used to qualitatively identify the type of glassy dynamics associated with a transition, is ≈ 0.0025 for T_G , within the range of a spin glass transition. As the existence of slow DC magnetization dynamics both below and above this feature suggests this glassy transition differs significantly from a classic freezing transition, we additionally probed $T_c = 175$ K. While the real part of the AC signal is dominated by the ferromagnetic signal and does not show a significant frequency dependence, an estimate of the Mydosh parameter using the frequency dependent χ'' signal is ≈ 0.006 , also consistent with a spin glass. This small difference between Mydosh parameters at T_c and T_G , as well as the qualitatively similar DC relaxation behavior below and above T_G , are thus consistent with identifying the 125 K transition as a second freezing transition, from a more dynamic glass to a stiffer glass phase interacting with the FM.”

We believe that these additions provide strong evidence for a spin glass phase.

In addition, the current measurements appear not complete to support a solid claim, particularly when magnetism of this material system might be complicate. To make a complete story and to understand the unusual EB phenomena better, several other measurements will be needed. For

example, extend the magnetization measurements (Fig. 3) to low temperature and compare with the Hall resistance (Fig. 1), magnetic field orientation dependence EB in magnetization measurements, etc.

As requested by the reviewer, we have added low temperature magnetization measurements showing EB to complement the Hall resistance data presented in Fig.1. This data is presented in Fig. 1(d) in the revised manuscript.

We have also included transport data to complement the magnetization measurements in Fig. 3. This data is now presented in Fig. 4(b). We would like to thank the reviewer for requiring this, as we feel this data set better explains the data set of the “zero-field-warming-up” showing the sharp feature at 125 K.

We have also performed additional orientation-dependent measurements. It should be expected that there is some anisotropy since this is a layered compound. The EB at 2 K is slightly stronger cooling in 45 degrees. We believe these anisotropies contain information pertaining to the microscopic nature of the interplay of the FM and frustration that drives the SG transition. However, this is not part of the scope of this work, which is focused on the report of the existence of exchange bias and the existence of a SG phase.

We include these results here:

Hall resistance as a function of magnetic field at a temperature of 2K, after cooling the sample in a nominal field of 1T applied to the sample at different angles. Once the sample reached 2K, the field is swept from between +1T and -1T and is repeated 5 times, resulting in a different H_C . The coercive fields on the opposite side are the same and therefore fall on top of each other in the plot.

When comparing the “relaxed” (i.e last loop) value of H_C . for the different cooldowns, the H_{EB} is slightly stronger for the 45-degree cooldown. This contains information pertaining to the microscopic nature of the spin glass and its interaction with the ferromagnet and is part of our efforts for a following work.

Further, for consistency the heat capacity should be performed under the same magnetic field as the magnetization in Figs. 5a-c.

The heat capacity under a magnetic field was performed to be consistent with the measurement of warming up in zero-field after cooling down in field. As such, a higher field was chosen than the one in the magnetization measurement. A review of Fig. 4 in the original manuscript (Fig. 3 in the revised one) reveals that the effect is only present for fields higher than 50 mT. We have added a clarification in the caption of the figure to explicitly convey that the field cooled heat capacity measurement is meant to complement the transport measurement in the previous figure and not the magnetization measurement.

We have repeated the measurement at the low field suggested by the reviewer to verify that this also does not show any feature at 125 K, and have included this measurement in the supplementary material.

Reviewer #2 (Remarks to the Author):

The manuscript focuses on the anomalous Hall effect in $\text{Co}_3\text{Sn}_2\text{S}_2$, a ferromagnetic Weyl semimetal which has been widely studied in the last two years. The manuscript reports on an exchange bias observed in the anomalous Hall effect which the authors use to support the claim, that in $\text{Co}_3\text{Sn}_2\text{S}_2$ there are two coexisting magnetic phases (ferromagnetism and (frozen) spin glass). They speculate that the coexistence of these two phases can give rise to an intrinsic (short range) exchange bias in the low temperature limit ($T < 125\text{K}$), where the spin glass phase is frozen. Similar results have been obtained by muon spin resonance suggesting a slightly different temperature window for the coexistence of an antiferromagnetic (not spin glass!) and ferromagnetic phase (90K-175K). The results in the present study do not corroborate this picture, as in this case one would expect to observe the exchange bias in this temperature window and not below.

To further underpin their findings and shine some light on the relevant mechanism they show the exchange bias to be present also in magnetization measurements in the low temperature region, while no exchange bias is present but an additional feature appears for $125\text{K} < T < 175\text{K}$. This, together with the exchange bias supports their claim of the two coexisting phases in this material also for the low temperature region, contradicting the findings of the muon spin resonance study.

As such, I believe this study is an important contribution to the ongoing discussion on the exact magnetic phase diagram/magnetic state of $\text{Co}_3\text{Sn}_2\text{S}_2$. I recommend the manuscript for publication in Nature Communications after the following questions have been addressed:

Major questions:

1. Usually, the proof for the presence of a spin glass phase are non-trivial time dependent magnetization effects. The authors state that no such time/frequency dependent changes were observed also in the spin glass phase, which they claim is because it is a dense spin glass and the time scales are too low.

However, in Ref. 12 a characteristic timescale of 1s is stated for the spin glass phase. How were the time dependent experiments performed (ac-susceptibility?). Any independent proof of the spin glass phase in the samples used for this study would extremely strengthen the point of the manuscript in my opinion.

We thank the reviewer for the positive review and recommendation for publication.

The reviewer is correct in that a spin glass phase should present non-trivial time dependent magnetization effect. New measurements we have performed indeed show such non-trivial time dependent, in agreement with Kassem *et al.* and we have removed the comment regarding time scales in the revised manuscript.

Before specifying these measurements, we would like to reiterate that though a spin glass phase is usually characterized by magnetization relaxation, in our case the spin glass coexists with a strong ferromagnetic order with a higher transition temperature. As such, all typical measurements of the spin glass would be corrections to the signal from the ferromagnetic phase. We therefore performed a different set of measurements that we feel still convey the non-trivial time dependent nature of the magnetism in $\text{Co}_3\text{Sn}_2\text{S}_2$:

- We have performed a magnetization relaxation measurement where we pause the magnetic field sweep during the hysteresis loop. By polarizing the ferromagnet at high negative fields and pausing at low positive fields we are in the region where the FM is softer and indeed these measurements show a magnetization relaxation time of several minutes. This measurement is presented in Fig. 6 (a) in the revised manuscript.
- AC susceptibility measurements were also added to the revised manuscript, showing a peak in the vicinity of 125K and a frequency dependence of the peak location. This measurement is presented in Fig. 6 (b,c) in the revised manuscript.
- In addition, a field sweep rate comparison at low temperatures (2K, 20K) shows that when sweeping the field at a rate of 20mT/sec the hysteresis loop area is slightly larger than when sweeping the field at a rate of 1mT/sec. This measurement was added to the supplementary materials, as we feel both the above measurements are sufficient to include in the main text.

2. In the study by Guguchia *et al.* (Ref. 11) the presence of an antiferromagnetic phase was shown to be present in the range from 90K to 175K. Is there any feature in the heat capacity down to 80K which might corroborate these findings? The authors show the heat capacity only down to 100K, so the temperature window should be extended down to at least 80K to strengthen their argument for the spin glass phase.

As per the reviewer's request, we have repeated the HC measurement down to 80K in order to include any possible feature that might be present in the heat capacity. The new data is now presented in Fig. 5(c) and does not show a feature at 90K.

3. The authors show the low field Hall measurement in Fig. 1&2, then show temperature dependent magnetization measurements in Fig. 3, how do the two measurements relate? Is the EB the same for magnetization and Hall measurements also for different temperatures? Is the spontaneous EB also visible? How does the magnitude of the anomalous Hall effect compare to the size of the step in Fig. 4 at 125K (rough estimate from other data shows 50% of maximum AHE amplitude, but comparison is hard [see minor question 2])? Is this something to expect from the frozen spin glass picture? Can the fraction of the $\text{Co}_3\text{Sn}_2\text{S}_2$ which is pinned be estimated from this?

These questions should be addressed by also showing the Hall measurements for different temperatures in a similar fashion as in Fig. 3

We have added a complementary transport measurement to Fig. 3 (Fig. 4 in the revised manuscript). This was a very helpful comment, as we feel that the transport data of the hysteresis loops does indeed help in the understanding of the data in Fig. 4 (Fig. 3 in the revised manuscript). We have also changed the order between these figures and have added the following paragraph to emphasize:

“The evolution of the magnetic hysteresis in R_{xy} as the temperature is increased through T_G sheds light on the 125 K anomaly presented in Fig. 3. When the sample is cooled in field $|\mu_0 H_{cool}| > 0.05$ T to low temperatures $T < T_G$ it is magnetized and retains this magnetization as the field is swept to zero. When subsequently warming up in zero applied field, the value of R_{xy} retains its magnetized value, until R_{xy} at zero field becomes significantly lower (about 80%) than its high field value at the same temperature.”

The more careful comparison of R_{xy} with the magnetization that the reviewer suggests is extremely interesting, and one we cannot presently answer in this manuscript. Comparing the magnetic hysteresis loops, it can be readily observed that while there is a net FM moment at zero field at temperatures around T_G , the anomalous Hall effect only retains a saturated zero field value below 125 K. We believe this is because the interplay between FM and SG opens a gap. This suggests an exciting direction would be to study whether this gap is associated with a quantum anomalous Hall effect, but this is of course beyond the scope of this work.

We have also included a complementary magnetization measurement in Fig. 1(d), showing the EB effect at low temperatures.

Minor questions:

1. Since the authors say they cannot use the a symmetrization due to the symmetric nature of the exchange bias, they subtract a geometric factor from the transverse data. As such, for being able to reproduce the data/evaluation I think it is crucial that also the longitudinal transport data (R_{xx}/ρ_{xx}) are shown to be able to judge possible artifacts introduced by this approach.

Longitudinal transport data for all relevant transport measurements were added to the supplementary material. For each, “non-mixed” both R_{xy} and R_{xx} are presented side by side.

2. To be able to compare the manuscript to other literature also regarding the effect magnitudes, the authors have to either convert their graphs to resistivities (not resistance) or give the sample size/contact spacings, so other scientists are able to convert the results to said units. Otherwise, it is very hard to compare the findings quantitatively.

The contact spacing information was add to the methods section, as requested.

3. Can the spontaneous EB also be related to domains?

The reviewer asks an interesting question. I would naively expect domains to be formed during the cooldown, and if there's a non-zero sum of opposite domains causing the SEB for that to be reflected in the initial value of the ZFC curve, and to not depend on p-type or n-type field sweeping protocol.

If the pinning is done by the domain walls, one would expect the SEB to vanish once the sample is completely polarized, which does not happen as is evident in the ZFC data in Fig. 1c (black curve). There, repeated sweeps – though some relaxation is present – show the same inclination of EB which, in the absence of cooling field, is spontaneous.

The SEB in this $\text{Co}_3\text{Sn}_2\text{S}_2$ no doubt comes for the unusual magnetic phases in the material, but it is unlikely in our opinion that the underlying cause is ferromagnetic domain walls.

Reviewer #3 (Remarks to the Author):

In the manuscript NCOMMS-19-23172, Lachman et al. reported the exchange bias effect observed in the ferromagnetic Weyl semimetal $\text{Co}_3\text{Sn}_2\text{S}_2$. Although the magnetic behaviors have been reported multiple times in literature, a clear picture of the magnetic order is absent. For this material, it is established that there is a strong magnetic anisotropy below T_c , which is known to be around 175 K. However, below this temperature, several studies have indicated the existence of a second magnetic transition and the possible existence of complex spin state or anomalous phase, for example, according to the result reported in the paper by Kassem et al. [ref. 12]. Indeed more experimental work is necessary to clarify the magnetic order and its origin. In the current work, by the Hall effect and magnetization measurements under field-cooling and zero-field cooling conditions, the authors report the signature of exchange bias effect, which from a different angle demonstrates the existence of the anomalous phase. This result seems to indicate the coexistence of the ferromagnetic phase and the anomalous phase. To reconcile the observed phenomena, the author proposed a spin glass transition below $T_G = 125$ K, and assume the FM onsets with frustrated AFM correlations, which then freezes at T_G into a dense spin glass. Since the provided evidences are quite limited in this manuscript, I do not think they are convincing to support the proposed physical picture. I do not recommend it as a publication in Nature Communications. Below I list a few suggestion to further improve the manuscript for a publication at a less selective journal.

We thank the reviewer for their comments. We would like to emphasize that in the revised manuscript we have added additional measurements we feel are more compelling evidence to the existence of a spin glass phase in $\text{Co}_3\text{Sn}_2\text{S}_2$. The revised manuscript now includes magnetic relaxation on a timescale of minutes in Fig. 6(a), and AC susceptibility measurements showing a peak in the vicinity of T_G with frequency dependence as expected from a spin glass. We have also added this paragraph to the revised text:

“AC susceptibility measurements around T_G presented in Fig. 6(b,c) reveal a small frequency dependence of the transition peak, indicating a glassy transition. The Mydosh parameter [15, 16], used to qualitatively identify the type of glassy dynamics associated with a transition, is ≈ 0.0025 for T_G , within the range of a spin glass transition. As the existence of slow DC magnetization dynamics both below and above this feature suggests this glassy transition differs significantly from a classic freezing transition, we additionally probed $T_c = 175$ K. While the real part of the AC signal is dominated by the ferromagnetic signal and does not show a significant frequency dependence, an estimate of the Mydosh parameter using the frequency dependent χ'' signal is ≈ 0.006 , also consistent with a spin glass. This small difference between Mydosh parameters at T_c and T_G , as well as the qualitatively similar DC relaxation behavior below and above T_G , are thus consistent with identifying the 125 K transition as a second freezing transition, from a more dynamic glass to a stiffer glass phase interacting with the FM.”

Even though our observation of exchange biased AHE is unambiguous, we agree with the reviewer that our arguments for the SG phase were not strong. However, the added data showing dynamic relaxation of magnetization, AC susceptibility feature and field-sweep-rate dependence of the hysteresis loop presented in the supplementary information, make a significantly stronger case for our claim that the additional phase underlying the EB effect is a spin glass.

1. The authors define the additional phase transition as spin-glass transition (TG) at 125 K. However, from ZFC magnetization curve (Fig. 5a), the transition anomaly seems to be around 140 K, while from the ZF magnetization curve (Fig. 5a), an anomalous temperature point seems to be around 160 K. Taking all the information together, the transition temperature at 125 K does not appear to be well defined.

Additionally, it is debatable to relate the sharp jump of Hall resistivity at TG (Fig. 4) to the anomalous magnetic transition. The sharp jump of the Hall resistivity should be directly related to the ferromagnetic domain dynamics, but it is not necessarily related to the spin-glass transition.

While T_G is marked in the magnetization shown Fig. 5, this is not the way it was determined. Rather the transition temperature for the anomalous phase was determined by the temperature in which the EB is no longer observed. We emphasize that this determination is unambiguous in the data of Fig. 4 in the revised manuscript (Fig. 3 in the original version). This temperature coincides with a visible peak in out-of-plane ZFC magnetization marked in Fig. 5(a) and a sharp rise in in-plane ZFC magnetization marked in Fig. 5(b). Note that magnetic signatures of SGs are always broad, since they arise from frozen disorder, and this is why we only use such data as a way to corroborate the approximate temperature scale of the transition.

In addition, our new AC susceptibility measurements also show a feature around 125K. These are presented in Fig. 6(b,c) in the revised manuscript.

These measurements and claim are in agreement with other papers we have cited (such as Kassem et al.), in which the anomaly temperature is identified at 125K.

2. The provided experimental data is not enough to support the existence of dense spin glass and antiferromagnetic correlation above TG. These existence of these two, however, are the core ingredients of the proposed physics behind.

We emphasize that one of the main points of the paper is the hitherto unreported finding of exchange-biases AHE. This observation can only be explained by an additional phase of either antiferromagnetic or spin glass nature coexisting with the known ferromagnetic phase.

We do not have evidence that there are AFM correlations at $T > T_G$, nor do we claim to. This was a suggestion to explain existing muon data in other studies.

As for the spin glass, which we do claim, we have added further measurements we feel provide compelling evidence to the existence of such a phase in $\text{Co}_3\text{Sn}_2\text{S}_2$ in the revised manuscript.

- We have performed a magnetization relaxation measurement where we pause the magnetic field sweep during the hysteresis loop. By polarizing the ferromagnet at high negative fields and pausing at low positive fields we are in the region where the FM is softer and indeed these measurements show a magnetization relaxation time of several minutes. This measurement is presented in Fig 6 (a). in the revised manuscript.
- AC susceptibility measurements were also added to the revised manuscript, showing a peak in the vicinity of 125K and a frequency dependence of the peak location. This measurement is presented in Fig 6 (b,c). in the revised manuscript.

- In addition, a field sweep rate comparison at low temperatures (2K, 20K) shows that when sweeping the field at a rate of 20mT/sec the hysteresis loop area is slightly larger than when sweeping the field at a rate of 1mT/sec. This measurement was added to the supplementary materials, as we feel both the above measurements are sufficient to include in the main text.

3. The information in the figure is not clearly conveyed. For example, The curves in fig. 1b, Fig. 2, Fig. 3, and Fig.4 seem to be vertically shifted for clarity, but the captions failed to indicate such information. Additionally, the temperature information for each MH is encouraged to be labeled, as there is enough space in the figure.

We thank the reviewer for catching the omitted clarification and have added the clarification for the above figures where the curves were indeed shifted for clarity.

We have also added the temperature information for each magnetization and transport measurement in the relevant figure (Fig. 4 in the revised manuscript).

4. The authors claimed to deduce the transition temperature of 125 K from Fig. 3a. This is also debatable. The temperature evolution of the MH curves is too subtle. More solid evidence is required.

We deduce the transition temperature of 125 K as this is the temperature where exchange bias is no longer observed. Originally, this temperature was deduced from Fig. 4 in the original manuscript, where the sharp feature in R_{xy} is observed in the warming up curves after cooling in field. We have changed the order of the figures (3 and 4) so that this is better conveyed. We have also added the explicit statement: "The EB is diminished at T_G , though the system is still magnetic."

We refer the reviewer to our response for comment no. 1, where we have laid out the rationale for identifying 125 K as the transition temperature, in agreement with other papers cited in our manuscript.

REVIEWERS' COMMENTS:

Reviewer #1 (Remarks to the Author):

In the revised manuscript the authors have addressed all issues raised in my previous comments. With the new data and discussions, the revised manuscript is ready to publish. I recommend the publication of this nice work.

Reviewer #3 (Remarks to the Author):

In updated manuscript NCOMMS-19-23172A, Lachman et al. significantly polished their work, by providing quite a few new experimental evidences to support their claim. This work now has seen a substantial improvement based on the feedback from reviewers. The story flow is much smoother, the data readability is better, and the data interpretation is appropriate based on the collected experimental evidences. The authors did a great job. I recommend it for a publication, once the authors clarify the following two minor concerns/issues:

1. Figure 4 shows the magnetization and transport measurements as a function of magnetic field at different temperatures around TG for samples cooled in a field of 0.5 T. However, I noticed one inconsistency issue by comparing the magnetization data with the Hall resistance data at 135 K, where the former one retains a plateau when the field is swept down and cross the zero field, while the latter one goes to almost zero at zero field. What is the origin of this inconsistency?
- 2, In Fig. S6, the amplitude of the SdH oscillations at 50 Tesla seems to have no temperature dependent decay behavior with increasing temperature. This behavior looks similar to that observed in the magnetic 2D material of GdTe₃ (Fig. 2D in arXiv:1903.03111v2). Can the authors comment on this point?

NCOMMS-19-23172A reviewers' comments

Reviewer #3 (Remarks to the Author):

1. Figure 4 shows the magnetization and transport measurements as a function of magnetic field at different temperatures around TG for samples cooled in a field of 0.5 T. However, I noticed one inconsistency issue by comparing the magnetization data with the Hall resistance data at 135 K, where the former one retains a plateau when the field is swept down and cross the zero field, while the latter one goes to almost zero at zero field. What is the origin of this inconsistency?

The reviewer is correct in identifying the inconsistency between transport and magnetization data. As the referee stated in their previous criticism, R_{xy} data is related to the magnetization of the system, but this relationship is not one-to-one. We believe this to be a very interesting feature of the data, and perhaps related to the difference between properties connected to symmetry breaking (the magnetic order parameter measure by M) and the properties connected to the topology (perhaps surface states or anomalous scattering in the bulk). It is possible that the onset of responses connected to each of these are not perfectly correlated. The explanation may also be more benign; for example, it may be that equilibrium (magnetization) and non-equilibrium response functions (transport) simply lag one-another in a highly degenerate landscape of a frustrated magnet.

As we have stated previously, the anomaly in R_{xy} only serves as a hint to the existence of the complex magnetism and the interplay between the glass and the ferromagnetic behavior that turns on at lower temperatures. The effect of this complex magnetism on the electronic transport is indeed an interesting and important question, and we are currently investigating this. However, although this remains an open question, it does not affect our main conclusions (namely the presence of exchange bias due to glassy behavior). We will state again that the magnetic feature based on which we have determined TG is present in all samples at 125K.

2, In Fig. S6, the amplitude of the SdH oscillations at 50 Tesla seems to have no temperature dependent decay behavior with increasing temperature. This behavior looks similar to that observed in the magnetic 2D material of GdTe3 (Fig. 2D in arXiv:1903.03111v2). Can the authors comment on this point?

We appreciate the reviewer's comment. The behavior referred to is intriguing, and merits a further investigation. A small caveat to that is that the range in question for GdTe3 is roughly 12K and the plateau ends at the Neel temperature. We are showing a range of only 1K. From the supplementary material of Liu et. al (ref 2 in our paper) it appears there is a reduction of said peak's amplitude at 5K. For Co3Sn2S2 the relevant temperatures for magnetism are much higher, nevertheless we agree that a further investigation would be interesting, albeit out of this paper's scope.